# Circulating Biomarkers for Monitoring Chemotherapy-Induced Cardiotoxicity in Children

**DOI:** 10.3390/pharmaceutics15122712

**Published:** 2023-11-30

**Authors:** Luigia Meo, Maria Savarese, Carmen Munno, Peppino Mirabelli, Pia Ragno, Ornella Leone, Mariaevelina Alfieri

**Affiliations:** 1Department of Chemistry and Biology, University of Salerno, Via Giovanni Paolo II, 132, 84084 Salerno, Italy; lmeo@unisa.it (L.M.); pragno@unisa.it (P.R.); 2Clinical Pathology, Santobono-Pausilipon Children’s Hospital, 80123 Naples, Italy; m.savarese@santobonopausilipon.it (M.S.); c.munno@santobonopausilipon.it (C.M.); o.leone@santobonopausilipon.it (O.L.); 3Clinical and Translational Research Unit, Santobono-Pausilipon Children’s Hospital, 80123 Naples, Italy; p.mirabelli@santobonopausilipon.it

**Keywords:** pediatric cancer, children, chemotherapy, cardiotoxicity, troponin, high-sensitivity-troponin, cardiac biomarkers, heart failure

## Abstract

Most commonly diagnosed cancer pathologies in the pediatric population comprise leukemias and cancers of the nervous system. The percentage of cancer survivors increased from approximatively 50% to 80% thanks to improvements in medical treatments and the introduction of new chemotherapies. However, as a consequence, heart disease has become the main cause of death in the children due to the cardiotoxicity induced by chemotherapy treatments. The use of different cardiovascular biomarkers, complementing data obtained from electrocardiogram, echocardiography cardiac imaging, and evaluation of clinical symptoms, is considered a routine in clinical diagnosis, prognosis, risk stratification, and differential diagnosis. Cardiac troponin and natriuretic peptides are the best-validated biomarkers broadly accepted in clinical practice for the diagnosis of acute coronary syndrome and heart failure, although many other biomarkers are used and several potential markers are currently under study and possibly will play a more prominent role in the future. Several studies have shown how the measurement of cardiac troponin (cTn) can be used for the early detection of heart damage in oncological patients treated with potentially cardiotoxic chemotherapeutic drugs. The advent of high sensitive methods (hs-cTnI or hs-cTnT) further improved the effectiveness of risk stratification and monitoring during treatment cycles.

## 1. Introduction

Cancer is the top disease-related cause of death for children and teens, although most cancers (99%) develop in adults. However, increasing research aims to discover new treatments for childhood cancers, greatly improving the overall survival rate, which has reached more than 80% for children. In the pre-chemotherapy era, the cure rate of childhood cancer was <25%, and most cancers were treated with surgery and radiation. With the advent of chemotherapy in the 1960s, and the development of novel therapies, a crescent increase in survival was observed. In the last years the development of next-generation sequencing technologies (NGS) and the advent of -omic sciences have shed light on molecular and genomic mechanisms underlying pediatric cancer disease, demonstrating that the genomic landscape is very various and, frequently, the etiology and tumorigenic mechanisms are quite different from that of common adult cancers [1,2,3].

Every year there are about 1400 diagnoses of cancer in pediatric population (Available online: https://apps.who.int/iris/handle/10665/347370, accessed on 29 July 2022; Available online: https://seer.cancer.gov, accessed on 29 July 2022) (Figure 1). However, despite the progress in the treatment of pediatric cancers based on the use of radiotherapy and poly-chemotherapy, several challenges remain, such as the management of severe acute and chronic toxicities caused by chemotherapies. More effective and less toxic treatments for pediatric cancers could be obtained by therapies that specifically target tumor cells, or inhibit oncogenic molecular aberrations as well as personalized chemotherapies through pharmacogenetics and precision medicin. Chemotherapy targets cells that are growing and dividing actively through cell cycle and especially cancer cells that grow faster than healthy cells. However, normal cells can be damaged by the chemotherapy as well, giving rise to several side effects. Although different for each person, side effects associated with chemotherapy administration comprise secondary malignancies, neurocognitive dysfunction, cardiopulmonary toxicity, impairment of kidney function, endocrinopathy, whose frequency and severity depend on several variables such as sex, age at diagnosis, and exposures to cumulative doses of specific treatment modalities [4]

Exposure to alkylating agents, anthracyclines, and epipodophyllotoxins, for treating hematologic malignancies, solid tumors and preconditioning regiments for hematopoietic stem cell transplantation (HSCT), has been associated with increased risk of developing secondary malignant neoplasms (SMN), histologically distinct malignancies, that may develop after completion of therapies for the primary tumor and remain the main cause of morbidity for pediatric patients [5]. Neurologic complications were observed in 33% of children with non-Central Nervous System (CNS) solid tumors. In fact, side effects of chemotherapy on central and peripheral nervous system are well-known and widely described and represent a significant source of morbidity in the treatment of cancer patients. CNS side effects are also found in patients subjected to new therapies, such as biologic therapy and immunotherapy; they occur during treatments or months/years later. Several symptoms and complications have been reported such as edema, seizures, fatigue, psychiatric disorders, and venous thromboembolism, that can earnestly impact the quality of life. Oxidative injuries from oxygen radicals generated by chemotherapy agents often result in acute or chronic cardiovascular complications. The cardiotoxicity of anthracyclines is widely known, but also alkylating agents (cyclophosphamide, ifosfamide, cisplatin, busulfan, mitomycin), vinca alkaloids, fluorouracil, cytarabine, amsacrine, asparaginase and the more recent agents as paclitaxel, trastuzumab, etoposideand teniposide, have also been associated with cardiotoxicity. Cardiac effects may occur during or immediately after treatment or months or years later after treatments including a wide spectrum of manifestations such as asymptomatic electrocardiographic abnormalities, blood pressure changes, arrhythmias, myocarditis, pericarditis, cardiac tamponade, acute myocardial infarction, cardiac failure, shock, and long-term cardiomyopathy. The assumption of anthracyclines (Doxorubicin and Daunorubicin) at doses higher than 300 mg/m^2^ results in symptomatic cardiotoxicity and is associated with high morbidity and mortality in children [6,7].

Cancer-therapy-related cardiac dysfunction (CTRCD) includes a spectrum of cardiac dysfunctions ranging from asymptomatic cardiac injury characterized only increased levels of cardiac biomarkers to clinical signs of heart failure. CTRD is less studied than in adults, although the toxic limited doses of chemotherapy agents are well known in children. In fact, doses of most chemotherapy agents in pediatric (but also in adult) population are established considering toxicity. The Children’s Oncology Group Chemotherapy Standardization Task force developed a method of dosing anticancer drugs in infants based on body surface area (BSA) dose banding incorporated in dosing tables for infants and children. These tables start from doses for children and infant with a BSA < 0.6 m^2^ and gradually switch from body weight based to BSA-based dosing [8,9]. Due to the increasing number of surviving children, CTRCD will continue to increase in the future flanked by the rapid evolution of the “cardio-oncology” aiming to detecting, monitoring, and treating cardiotoxic side effects of cancer therapy [10]. The National Comprehensive Cancer Network’s “Adolescents and Young Adults with Cancer” and The Children’s Oncology Group’s “Long-Term Follow-Up Guidelines for Survivors of Childhood, Adolescent, and Young”, have established guidelines containing detailed informations to establish programs for childhood cancer survivors (Available online: https://cancerprogressreport.aacr.org/wp-content/uploads/sites/2/2022/09/AACR_CPR_2022.pdf, accessed on 26 August 2023; Available online: http://survivorshipguidelines.org, accessed on 26 August 2023). Prevention of cardiotoxicity may be achieved by screening for risk factors, monitoring for symptoms and events during treatments, electrocardiographic and echocardiographic studies, angiography, measurements of biochemical markers of myocardial injury. A second level of prevention aims to minimize progression of heart dysfunction by monitoring cardiac function during and after cancer therapy, using cardioprotectants reducing oxidative stress during chemotherapy, using analogs or formulations having milder cardiotoxicity, altering the dose, schedule or approach to drug delivery.

In this review, we recapitulate the effects of different chemotherapy drugs on heart function and of main biomarkers used to detect and follow cardiotoxicity underlining the key role of the high-sensitivity troponin as a biomarker for monitoring chemotherapy-induced cardiotoxicity in pediatric patients.

## 2. Chemotherapy-Induced Cardiotoxicity in Children

Knowledge of the pediatric heart, compared to the adult heart, is essential to address the issue of chemotherapy-related cardiotoxicity. Long term cardiotoxicity is observed especially in pediatric patients who experiences relapses and second line treatments [11]. The fetal and neonatal heart express high levels of cyclin-dependent kinases and cyclins, positive cell cycle regulators, not expressed in the adult heart [12]. Moreover, in neonatal cardiomyocytes, the telomerase activity promotes S-phase entry and suppresses cyclin-dependent kinase inhibitors thus inducing cell proliferation [13].

Anthracyclines are the chemotherapy drugs mostly associated with cardiotoxicity. However, cardiac complications could be associated also with other chemotherapy agents under trial or commonly used for the treatment of pediatric cancers, such as paclitaxel, tyrosine kinase inhibitors and alkylating agents [14].

### 2.1. Anthracyclines

Anthracyclines are chemotherapeutic drugs acting by targeting both the isoenzymes of topoisomerase II (TOP2) (TOP2A and TOP2B) and stabilizing the TOP2 DNA complex. Disruption of the TOP2B on mice and human embryonic stem cells prevents the cardiotoxicity associated with anthracyclines demonstrating that TOP2B is responsible for the cardiotoxicity associated with anthracyclines [15,16]. Doxorubicin, the most commonly used anthracycline, act inducing lipid peroxidation at the cell and mitochondrial membranes by forming complexes with Fe^2+^, thus resulting in apoptosis, mitochondrial DNA damage producing reactive oxygen species (ROS) [17,18] whose persistence could lead to late CTRCD. Specifically, anthracycline is a lipophilic molecule that easily diffuses through the cell membrane, reaching the inner mitochondrial membrane. It acts by binding cardiolipin and converting its quinone into semiquinone thus realizing free radicals. This mechanism induces the production of ROS that contribute to lipid peroxidation, cell membrane damage and the release of intracellular proteins such as lactate dehydrogenase (LDH) and cardiac troponins [19,20]. The use of the iron chelator Dexrazoxane, in a preclinical study in mice with upregulated mitochondrial iron exporters, reduces the generation of ROS in mitochondria providing cardioprotection from anthracycline treatment without reducing its anticancer efficacy, thus mitigating CTRCD [21,22]. Dexrazoxane has been approved by FDA as a cardioprotective agent and has been employed to improve cardiac toxicity observed in anthracycline-treated oncologic patients, such as breast cancer patients or small-cell lung cancer and adult patients with soft tissue sarcomas [23]. This drug is administered intravenously in combination with the anthracycline to reduce the incidence of cardiac complications. Dexrazoxane can also be used to treat tissue damage caused by extravasation of anthracyclines during their administration [24]. However, since Dexrazoxane does not wholly eliminate the risk of anthracycline- induced cardiotoxicity, it is critical to verify cardiac function before and during therapy to monitor the left ventricular ejection fraction (LVEF) [23].

#### Genetic Variants Associated with Anthracycline Sensitivity

Several studies identified genetic variants associated with Anthracycline sensitivity in pediatric oncologic patients. A number of single nucleotide polymorphism (SNP) variations in specific genes, resulting in changes in signal transduction, cell cycle regulation, mitochondrial function and cellular transport, were identified (Table 1). These variants are responsible for the anthracycline-associated cardiotoxicity (ACT). The polymorphism rs1056892 in carbonyl reductase (CBR3) is related to a tripled risk of ACT [21]. The polymorphism rs3743527 within the 3′ Untranslated Region (3′ UTR) of the ATP binding cassette subfamily C member 1 (ABCC1) gene in children affected by Acute Lymphoblastic Leukemia (ALL) decreased the ABCC1 expression, through a posttranscriptional mechanism, promoting ACT [25]. The SNP rs10426377 of sulfotransferase family cytosolic member 2B1 (SULT2B1), which increases the solubility of drugs in water and promotes renal excretion, is also associated with ACT in men, probably decreasing the excretion of anthracycline through kidneys [26,27]. The SNP rs13058338 in the RAC2 subunit of NADPH oxidase is strongly related with ACT susceptibility. Indeed, NADPH oxidase knockout mice were protected from heart failure induced by doxorubicin; further NADPH oxidase inhibitors showed a reduction in the damaged cardiomyocytes after anthracycline exposure. Altogether, these studies confirm that the NADPH oxidase, and thus SNP rs13058338, plays a key role in the ACT [28]. The SNP rs10836235 in catalase (CAT), that converts hydrogen peroxide into water preventing the conversion to hydroxyl free radical, increases expression of CAT and is associated with ACT resistance [29]. The glutathione S transferase (GSTP1) conjugates free glutathione to free radical chemotherapeutic drug metabolites, thus preventing damaging interactions with molecules like DNA, protein and lipids. The SNP rs1695 of GSTP1 leads to a reduced activity of GSTP1 and, consequently, to ACT susceptibility [30]. Increased risk and severity of ACT was also associated to the missense mutation rs12468485 in the G protein-coupled receptor 35 (GPCR35) [31]. Hyaluronan synthase-3 (HAS3) is responsible of the synthesis of the glycosaminoglycan Hyaluronan, that serves as scaffolds during tissue remodeling. The SNP rs2232228 in HAS3 was reported to influence the risk of ACT [32]. The SNP rs1786814 in the CUGBP Elav-like family member 4 (CELF4) is associated to the propensity to develop cardiomyopathy post-anthracyclines in childhood cancer survivors [33]. CELF4 is an RNA binding protein that regulates the alternative splicing of the transcript of the TNNT2 gene encoding for cardiac troponin T, a component of the thin filaments of sarcomeres [33]. The embryonic heart predominantly expresses the variants of cardiac troponin T with an alternative exon 5, which, by contrast, is strongly down-regulated in adults. The CELF4 variant shows a lower affinity to the TNNT2 in the adult heart, impairing the contractility and in some cases LV ejection fraction [34]. Magdy and colleagues demonstrated that primary cardiomyocytes recapitulated the cardio-protective effect of the SLC28A3 locus and that SLC28A3 expression modulated the severity of anthracycline-induced cardiotoxicity [35]. A novel cardio-protective SNP, rs11140490, identified in the SLC28A3 locus, exerted its effect modulating the expression of the antisense long noncoding RNA SLC28A3-AS1 overlapping SLC28A3 locus, thus impairing the expression of doxorubicin–related genes like *SLC28A3*, and eventually protecting patients from ACT [35]. The SNP rs2229774, identified in retinoic acid receptor-γ (RARG) that binds the TOP2B promoter, was associated with an increased risk of anthracycline-induced cardiotoxicity. This nucleotide variation induces a decreased repression of the TOP2B, thus resulting in increased cardiomyocyte death [17]. In fact, human induced pluripotent stem cell-derived cardiomyocytes (hiPSC-CMs) derived from patients with rs2229774 were more sensitive to doxorubicin. This variant acts through the suppression of TOP2B expression and the activation of the cardio-protective extracellular regulated kinase (ERK) pathway [36]. The Canadian Pharmacogenomics Network for Drug Safety (CPNDS) identified, several genetic variants playing a significant role in the risk of developing ACT. In particular, three variants emerged as consistently associated with ACT, including the previously cited variant RARG (rs2229774, G>A), the variant UGT1A6 (rs17863783, G>T) and the protective variant in SLC28A3 (rs7853758, G>A). The UGT1A6 (rs17863783, G>T) variant reduces the UGT1A6 glucuronidation activity, that causes the accumulation of cardiotoxic metabolites resulting in an increased risk of developing ACT. The SLC28A3 (rs7853758, G>A) reduces the expression of the SLC28A3 gene encoding for a drug transporter of anthracyclines thus impairing the influx of anthracyclines in cardiac cells, producing a protective effect [37]. A missense variant (p.Thr253Met, c.758C>T variant) of the gene GPR35, identified using exome array data, was associated with an higher risk of chronic anthracycline-induced cardiotoxicity and more severe cardiac manifestations at low anthracycline doses [38]. A summary of identified genetic variants associated with Antracycline-related ACT is presented in Table 1.

### 2.2. Taxanes

Taxanes, diterpenes originally isolated from TAXUS, induce cardiotoxic events in 3–20% of patients [39]. Among taxanes, paclitaxel exacerbates anthracycline-induced toxicity increasing heart failure (HF) events [40] and histo-pathological alterations of cardiac tissue, with extensive necrosis [41]. Paclitaxel is an alkaloid extracted from vinca, acts by interrupting microtubule formation. It targets tubulin and stabilizes the microtubule polymer, protecting it from disassembly, thus impairing the metaphase spindle configuration and the progression of mitosis. Paclitaxel is highly effective in the treatment of recurrent childhood brain tumors and is becoming a prospective management option for pediatric tumors [42]. Another study demonstrated the effectiveness of paclitaxel treatment on children affected by Kaposi sarcoma [43]. Perez-Somarriba and colleagues observed a strong response with normalization of tumor markers in three patients showing relapsed intracranial NGGCT, after treatment with gemcitabine, paclitaxel, and oxaliplatin [44]. However, although taxanes have demonstrated high efficacy in adults affected by solid tumors and in pediatric solid tumor in vitro models, their use in children has been restricted by dose-limiting toxicity especially due to the solvent formulation of these drugs [45,46], so these drugs are not yet introduced in clinical practice for pediatric patients.

However, it has a cardiotoxic effect in the adult population such as bradycardia, cardiac ischemia, atrioventricular block, ventricular arrhythmias, probably due to the fact that the disruption of microtubular functions can definitely impact on subcellular organelle, indirectly interfering with myocardial functioning [47]. However, the cellular and molecular mechanisms underlying taxane-induced cardiotoxicity have not been fully elucidated.

### 2.3. Tyrosine Kinase Inhibitors

Tyrosine kinase inhibitors (TKIs) block the enzymes tyrosine kinases. Tyrosine kinases mediate the activation of growth signals in cells, thus impairing cell proliferation.

In particular, TKIs act by interfering with the ATP binding site of the tyrosine kinase or blocking the ligand-binding site impairing its function. Examples of TKIs include axitinib, dasatinib, erlotinib, imatinib, nilotinib, pazopanib, sunitinib. Used to treat cancers in children such as chronic myeloid leukemia (CML), acute myeloid leukemia (AML), acute lynphoblastic leukemia (ALL), gastrointestinal stromal tumor (GIST), neuroblastoma, renal cell carcinoma, etc. [48]. These drugs activate the endoplasmic reticulum (ER) stress response; the prolonged activation can promote cell death pathways that finally lead to apoptotic and necrotic death of the cardiac tissue.

### 2.4. Alkylating Agents

Alkylating agents, such as Cisplatin and Carboplatin, commonly used during bone marrow transplantation in children, can induce late-onset CTRCD [49]. These drugs cause an increased ROS production and direct injury to the endothelium causing extravasation of blood in the myocardium, deposition of fibrin in the interstitium, and capillary microthrombi [50]. Cardiomyocyte apoptosis, inflammation, endothelial dysfunction, calcium dysregulation, and mitochondrial damage, promoting hypertension, cardiomyopathy, myocardial infarction, arrhythmias, and heart failure (HF) have been observed in patients treated with Cyclophosphamide [51]. 5-Fluorouracil (5-FU) has been associated with the acute coronary syndrome, probably due to a decrease in nitric oxide locally, causing coronary artery spasm and vasoconstriction, microvascular thrombus, and finally myocardial ischemia [52].

### 2.5. Radiation Therapy

The 10–30% of patients receiving radiation therapy (RT) develop cardiac complications [53]. The effect of radiation onto water molecules in the cell produces ROS thus damaging mitochondria and DNA in myocytes thus promoting myocardial fibrosis. Cardiotoxicity severity depends on the total radiation dose and the volume of the heart exposed and its incidence increases by 60% for every 1-Gy increase in mediastinal radiation dose [54]. Further, cardiotoxicity is more commonly observed in patients who previously received anthracycline-based chemotherapy or showing cardiac risk factors. Radiation also causes endothelial dysfunction, resulting in intravascular thrombosis, myocardial ischemia, and interstitial fibrosis [55].

## 3. Cardiac Biomarkers

As stated by the American College of Cardiology and the European society of Cardiology, cardiac biomarkers are central in the diagnosis of major cardiovascular diseases. A good cardiac biomarker should be highly expressed in the cardiac tissue, show sensitivity and specificity and early detectable in the blood after the onset of clinical symptoms, such as chest pain [56,57,58]. There are many cardiac biomarkers, classified into different patho-physiologic groups, including myocardial ischemia or necrosis, inflammation, hemodynamics, angiogenesis, atherosclerosis, or plaque instability [59].

Aspartate transaminase, a not cardio-specific enzyme, was the first biomarker used for the diagnosis of Acute Myocardial Infarction (AMI) in the 1960s [60]. For this reason, later different other enzymes with increased sensitivity and specificity, such as LDH, creatine kinase (CK), CK isoenzyme MB (CK-MB), and myoglobin, were employed in the diagnostic algorithm, thus improving the diagnosis accuracy [61,62]. In the 1990s, a sensitive radio-immunoassay was developed to detect circulating troponin, for the biochemical diagnosis of AMI, with a sensitivity near to 100% in detection of AMI at 6–12 h after the onset of chest pain. Cardiac troponins T and I levels increase early after the onset of AMI, reaching a peak in the serum after 12–48 h and remaining elevated for 4–10 days [63]. In 2007–2010, a high-sensitivity assay (hs-cTn) became the gold standard in the diagnostic algorithm in the last universal definition of MI. The most largely employed cardiac biomarkers are troponins in the diagnosis of AMI and the natriuretic peptides in the diagnosis and prognosis of heart failure. Elevated cardiac troponin is the main criterion with high sensitivity and specificity, for the diagnosis of AMI in the presence of ischaemia signs while BNP and its N-terminal fragment proBNP (NT-proBNP) are considered markers with high sensitivity but low specificity for HF [64].

Cardiac troponin (cTn) is a component of the thin filament playing a key role in the muscle contraction by mediating the interaction between actin and myosin.

Three similar isoforms have been identified: the isoform C (cTnC) binds Calcium ions; the isoform I (cTnI) inhibits the ATPase activity of actomyosin; and isoform T (cTnT), interacts with actomyosin. Plasmatic levels of cTn are elevated also in acute or chronic heart failure, myocarditis, Takotsubo cardiomyopathy, atrial fibrillation, aortic dissection and stroke. cTn is released in the blood consequently to cell turnover, myocyte apoptosis or necrosis. It has been demonstrated that in response to ischemia without necrosis, membranous blebs release cTn. Isoforms cTnI and cTnT, expressed only in the cardiac muscle (contrary, the isoform C is expressed also in the skeletal muscle), are the most specific markers for acute coronary syndromes among the isoforms, and their plasmatic increase are considered the “gold standard” in AMI diagnosis [65].

In various studies, the diagnostic value of cTn after the treatment with cardiotoxic chemotherapy was investigated. Lipshultz and colleagues investigated cTnT blood levels in 51 sampled patients (median = 5.7 years) subjected to surgical cardiovascular (*n* = 19) or non-cardiovascular (*n* = 17) treatment or who took doxorubicin for acute lymphoblastic leukemia (ALL) (*n* = 15). They demonstrated that increased levels of cTnT in the serum of children corresponded to a more severe myocardial damage and predicted consequent subclinical and clinical cardiac morbidity and mortality [66].

However, another study showed that measurement of cTnT within 24 h after assumption of chemotherapy drug did not show high sensitivity to identify patients with following subclinical cardiotoxicity. cTnT levels were measured at three time points in 163 samples from 38 children, 24 h after chemotherapy cycles. In this study, increased levels of cTnT (>0.010 ng/mL) was observed in 6 samples from 3 patients. Finally, at the end of chemotherapy cycles, 7 out of the 38 patients showed LV dysfunction and only 1 of these 7 showed increased cTnT levels, while 2 children with higher cTnT levels did not show LV dysfunction up to 7 months after the cTnT measurement [67].

From the advent of high-sensitivity cardiac troponin (hs-cTn) assays, attention was more focused on the predictive value of cTn for cardiovascular disease beside its diagnostic use for acute coronary syndrome [68]. Actually, in most of worldwide laboratories the hs-cTn assays have been recognized and recommended by European Society of Cardiology Guidelines for rapid rule-in/rule-out of myocardial infarctions. These assays are used in the early evaluation of the typical chest pain and also for prognosis, to predict the risk of long-term events and mortality, not only in cardiovascular diseases but also in different pathologies. Compared to classical assays, hsTn assays are able to detect troponins at a much lower concentration, with high precision, with small coefficient of variation even at the 99th percentile, in the reference population, thus allowing a faster recognition of AMI (rule-in/rule-out) [69]. It is important to underline that there is a variability of troponin levels also in healthy patients, for example men show higher levels than women, there is a circadian variability (lightly higher in the morning). In addition, there is also a correlation with age and a variability in patients affected by diseases such as chronic kidney disease and diabetes [70].

The methods for the detection of hsTn are mainly immunochemical methods, such as enzyme linked immunoassay (ELISA), immunofluorescence assay, radioimmunoassay (RIA), and immune-chemiluminescence assay that share similar bases: an immunological phase, in which there is a specific interaction between an anti-cTn antibody and a cTn N-terminal amino acid antigen; a second phase characterized by either an enzymatic reaction or another antibody-antigen reaction, based on the method chosen; the final phase varies depending on the detection method employed: a spectrophotometer measuring color intensity for ELISA a radiometer detecting radionuclides emissions for RIA and, a fluorometer detecting fluorophores for immunofluorescence. The intensity of the signal is directly proportional to the amount of troponin molecules, allowing their quantification. 

A comprehensive reference table of hsTn methods available either on automated laboratory platforms or as point of care (POC) systems is regularly updated by the IFCC Committee on Clinical Applications of Cardiac Bio Markers (C-CB) and it is publicly accessible on the IFCC website [71]. For the purpose of this review, we focused on POC hsTn methods, which are amenable to bedside testing and are suitable for pediatric patients as they require small volume whole blood samples. As reported by the IFCC Committee on Clinical Applications of Cardiac Bio-Markers (C-CB), only three analytical systems for the measurement of cardiac troponin respond to both the definition of high sensitivity assays [72] and the most recent definition given by IFCC of point of care devices [73]: PATHFAST hs-cTnI (LSI Medience, Tokio, Japan), Quidel TriageTrue High Sensitivity Troponin I Test (Quidel, CA, USA) and Siemens Atellica VTLi hs-cTnI (Siemens Healthcare Diagnostics Inc., Erlangen, Germany). Their analytical and instrument characteristics are summarized in the Table 2.

BNPs belongs to the group of Natriuretic peptides (NPs), composed of three peptides with similar structures: the atrial natriuretic peptide (ANP), the B-type (or brain) natriuretic peptide (BNP), and the C-type natriuretic peptide (CNP). BNPs play a key role in the maintenance of the cardiovascular homeostasis, regulating volume and pressure overload. Circulating BNP and the N-terminal pro-B-type natriuretic peptide (NT-proBNP) increase their levels consequently to increased wall stretching due to volume or load stress in HF caused by systolic and/or diastolic dysfunction, valvular heart disease, left ventricular hypertrophy, ischemia, or a combination of these factors.

BNP and inactive NT-proBNP derive from the cleavage of the precursor proBNP and are then secreted in equal concentrations in the blood, where the BNP can bind to NP receptors (NPRs) thus activating the intracellular cGMP signaling pathways to decrease the volume or the pressure overload. BNP is mainly eliminated by degradation mediated by endopeptidases and in part through and renal excretion and the uptake by NPR [75].

Further, the levels of BNP or NT-proBNP are definitely helpful for risk stratification and management of patients with suspected heart failure. In fact, their decreasing levels indicates effective management strategies [76].

A study conducted on samples of 2525 patients three days after the onset of ischemic symptoms showed that measurement of the BNP could provide predictive information on the risk of new or recurrent MI episodes [77].

For an appropriate diagnosis, risk stratification, and management of patients affected by cardiovascular diseases, other biomarkers can be evaluated in routine clinical practice. Among these, C-Reactive Protein (CRP) and Interleukin-6 (IL-6) are both significantly upregulated in acute coronary syndrome [78]. CRP, most widely used as inflammatory marker in routine clinical practice, whose levels have been observed to significantly increase in ACS patients as compared to the patients without a story of ischemic cardiomyopathy [79].

Copeptin, is a peptide deriving from the C-terminus of the vasopressin prohormone. It is released with arginine vasopressin (AVP) within 0–4 h after the onset of cardiac symptoms [80] improving efficacy in combination with troponin levels measurement for early rule-out of AMI [81,82,83,84]. However, it is as a nonspecific prognostic marker, since its level is also influenced by other pathologic conditions also such as renal disease and lower respiratory tract infections [85].

Among emerging biomarkers still under study there are some cytokines such as Interleukin 6 (IL-6) and Interleukin 37 (IL-37) and a member of lectin family, Galectin-3 (Gal-3).

IL-6, as well as CRP, is an inflammatory biomarker showing high expression levels in MI induced by trans-coronary ablation of septal hypertrophy [86,87]. IL-6 is also related to adverse cardiac events [88]. These observations suggest the diagnostic value and the potential use as therapeutic target in unstable ischemic heart disease [89]

IL-37 is an anti-inflammatory cytokine belonging to the IL-1 ligand family [90] that plays a crucial role in innate immunity and adaptive immunity. It inhibits the secretion of cytokines IL-1*β*, IL-6, and TNF-*α* in monocytes, macrophages, dendritic cells, and epithelial cells [91] a is involved in the onset and development of chronic inflammation and autoimmune diseases such as systemic lupus erythematosus, diabetes and rheumatoid arthritis [92,93]. As demonstrated by clinical and and pre-clinical studies in animal models, IL-37 participates in atherosclerotic disease and its levels are highly upregulated in acute coronary syndrome in which increased IL-37 levels are correlated with poor outcomes [94,95].

Gal-3 is involved in inflammation processes and its levels increase in cardiac injury [96]. It is a member of the lectin family; it is related to left ventricular dilation and contributes in the prediction of the outcome and the follow up of patients with both acute or chronic HF [97,98].

In the last decade, potential prognostic value of lnRNA and miRNA emerged as new and sensitive biomarkers of myocardial infarction probably acting as molecular sponge [99,100,101,102,103,104,105].

Adachi and colleagues demonstrated that significant increased levels of circulating miRNA-208b and miRNA-499, specifically expressed in cardiomyocytes, have been detected in AMI patients compared with the health control group [106]. Serum levels of miRNA-1 and miRNA-133a, muscle-specific microRNAs, are involved in the regulation of cardiac hypertrophy; their serum levels resulted increased in a group of patients with unselected AMI. In particular, it has been shown that miRNA-133a detected in the serum can be emplyed as a marker for cardiomyocyte death [107]. The long noncoding RNA LIPCAR is considered a marker in patients with ST-segment elevation MI and can contribute to predict survival in patients with heart failure [108,109]. lnRNA HOTAIR is an essential mediator of acute myocardial infarction [110].

## 4. Role of hs-cTn in the Diagnosis and Monitoring of Chemotherapy-Induced Cardiotoxicity in Pediatric Population

The European Society of Cardiology (ESC) position paper on cancer therapy and cardiovascular toxicity (2016) recommends the use of high-sensitivity troponins to predict left ventricular (LV) dysfunction in patients undergoing chemotherapy [111]. The 2020 European Society for Medical Oncology (ESMO) consensus cites the importance of the measurement of hs-cTn in detecting or predicting cardiovascular toxicity, during oncologic therapy [112]. In addition, Recent Clinical Practice Guidelines in Cardio-Oncology suggested that myocardial biomarkers, are considered diagnostic indicators of CTRCD [113]. A number of studies demonstrated that hs-cTn could be used as a marker for early monitoring of CTRCD in chemo-radiotherapy-treated patients [114,115,116,117,118,119,120,121]. In general, an increase in Tn concentration by 3–5 ng/L, measured by the hs-cTn corresponds to 10–20 mg of myocardial tissue necrosis, which could not be detected by cardiac imaging techniques [122].

However, so far, reference values for pediatric population have not been established thus limiting the use of hs-cTn dosage for the diagnosis and managements of cardiac diseases in children. As demonstrated by some studies hs-cTnI and hs-cTnT values are generally more elevated in infants and children if compared to those observed in adults [123,124,125,126,127]; moreover, boys have usually higher values than girls confirming the trend already reported for the conventional methods [128,129,130]. In particular, values of hsTn are highest in the first month of life, then rapidly decline in the next six months and finally continue to decline slower to reach a plateau during adolescence. Some studies demonstrated the increase of hsTn values in healthy adolescents after athletic training, returning to baseline values in 24–48 h [131,132,133,134,135,136]. Although based on few studies, increased levels of hsTn are considered a marker of myocardial injury also in pediatric population [137]. The incidence of myocardial infarction is much lower than in adults but myocardial injury can be found in neonates, infants and children affected by bronchiolitis, sepsis, and has recently been observed in pediatric patients with SARS-CoV-2 [138,139,140].

In order to shed light on the relevance in using hsTnI method to detect TnI in children patients, we report here some data that we obtained in our hospital, presented in the 27° Annual Symposium ELAS-Italia, Ligand Assay (November 2022, Bologna, Italy). 46 plasma EDTA samples coming from 15 patients undergoing chemotherapy (age: 7 months–16 years) were analyzed retrospectively, with the high-sensitive hsTnI method following manufacturer instructions (Triage MeterPro/TriageTrue hs-cTnI Assay). Samples were selected as their cTn values were below the lower measuring limit of the cTnI method (0.05–30 ng/mL). Results were interpreted based on the overall 99th percentile URL reported for the high-sensitive method (20.5 ng/L). Out of the 49 samples resulting in values <0.05 ng/mL (<50 ng/L) with the cTnI method, 28 were below the overall 99th percentile URL of the high-sensitive method (<20.5 ng/L), 18 were above it (≥20.5 ng/L). Out of these 18 samples, 9 resulted in hs-cTnI values above 50 ng/L. These results suggest that the measurement of hs-cTnI granted us a higher level of analytical detail during the monitoring of chemotherapy-induced cardiotoxicity. Not only it was possible to observe values below the threshold of the cTnI traditional method, but in some cases hs-cTnI levels were found to be above it (up to 154 ng/L), due to the considerably higher precision of the new method. Moreover, in some patients, sequential hs-cTnI measurement revealed troponin oscillations following treatment, opening for its routine use in monitoring potential cardiac damage by chemotherapy agents (Figure 2) [141].

Finally, the lack of reference values in the pediatric population for all commercially available hs-cTn assays, a gap that we hope will be bridged soon, does not hinder their use for the present application, as chemotherapy-induced cardiac damage should always be monitored following measurement of individual basal values and evaluation of relative variations in the concentration of the biomarker.

## 5. Conclusions and Future Perspectives

Cardiomyopathy and heart failure are the main causes of death in cancer survivors, especially in children. For this reason, its early detection in cancer survivors is crucial for the prevention of long-term cardiovascular morbidity. Thus, circulating biomarkers are important tools to provide prognostic prediction and to guide cardio-protective therapy during chemotherapy treatment in oncologic patients [142]. Compared to adults, cancer-surviving children are more susceptible to the cardiotoxic effects of anthracycline due to both the life expectancy length and a deficient potential for myocardial growth to compensate for early damage and somatic development. In some survivor ALL doxorubicin-treated children, it has been observed thinner LV walls, increased afterload, and depressed contractility, progressively leading to morbidity or mortality. Early identification of cardiac damage in children after cardiotoxic therapy is crucial to improve clinical practice and benefiting patient care. Finally implementation of preventive measures can limit later myocardial damage. However, so far, no parameters have been identified to predict the possible development of cardiac dysfunction or HF in children.High sensitivity troponin is a very promising tool for early detection of CTRCD, but further prospective studies, particularly in childhood, but also in adulthood are required for its fully entry in clinical practice. Finally, the genetic can explain why some patients develop cardiotoxicity while others with same risk factors do not. This means that, in the future, pharmaco-genetic testing and personalized chemotherapy could be very effective in limiting CTRCD.

## Figures and Tables

**Figure 1 pharmaceutics-15-02712-f001:**
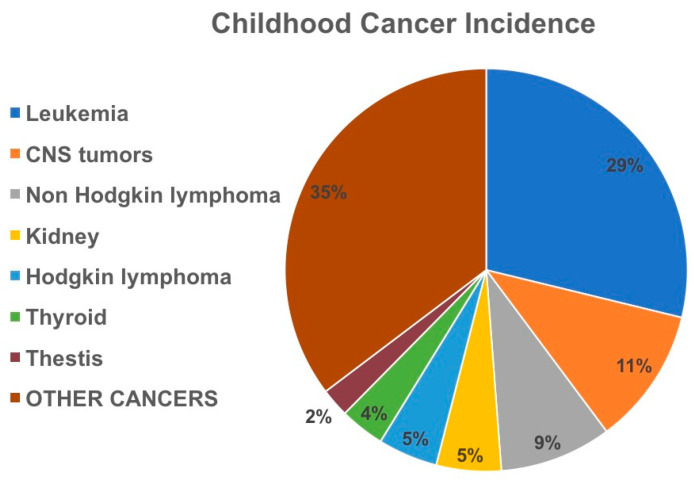
Frequency of pediatric cancers in pediatric population (0–19 years), calculated from Global Cancer observatory, accessed on 1 September 2023 (https://gco.iarc.fr/today/home).

**Figure 2 pharmaceutics-15-02712-f002:**
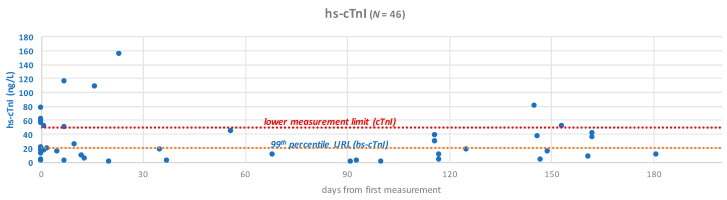
Overall results of hsTnI obtained from 46 children [141].

**Table 1 pharmaceutics-15-02712-t001:** Genetic variants associated with Anthracycline sensitivity in pediatric oncologic patients.

SNP	Locus	Effects	Reference
rs2229774	RARG	increased cardiomyocyte death	Aminkeng et al., 2015 [17]
rs1056892	CBR3	tripled risk of ACT	Blanco et al., 2012 [21]
rs3743527	3′ UTR of ABCC1	ACT	Semsei et al., 2012 [25]
rs10426377	SULT2B1	ACT	Visscher et al., 2012 [26] Visscher et al., 2013 [27]
rs13058338	RAC2 subunit of NADPH	ACT susceptibility	Zhao et al., 2010 [28]
rs10836235	CAT	ACT resistance	Rajić et al., 2009 [29]
rs1695	GSTP1	susceptibility to ACT	Windsor et al., 2012 [30]
rs12468485	GPCR35	increased risk and severity of ACT	Min et al., 2010 [31]
rs2232228	HAS3	manipulate the risk of ACT	Wang et al., 2014 [32]
rs1786814	CELF4	increased cardiomyopathy	Wang et al., 2016 [33]
rs11140490	SLC28A3 locus	cardio-protective effect	Magdy et al., 2022 [35]
rs2229774	RARG	increased risk of ACT	Magdy et al., 2022 [35]
rs17863783	UGT1A6	increased risk of ACT	Loucks et al., 2022 [37]
rs7853758	SLC28A3	cardio-protective effect	Loucks et al., 2022 [37]
p.Thr253Met, c.758C>T variant	GPR35	increased risk of ACT	Ruiz-Pinto et al., 2017 [38]

**Table 2 pharmaceutics-15-02712-t002:** Analytical and instrument characteristics of high-sensitivity cTn assay.

Characteristic	PATHFAST hs-cTnI	Quidel TriageTrue hs-cTnI	Siemens Atellica VTLi hs-cTnI
LoB	1.23 ng/L	0.4 ng/L (plasma) 0.5–0.8 ng/L (whole blood)	0.55 ng/L
LoD	2.33 ng/L	0.7–1.6 ng/L (plasma) 1.5–1.9 ng/L (whole blood)	1.2 ng/L (plasma) 1.6 ng/L (whole blood)
% CV at 99th Percentile	6.1%	5.0–5.9% at 21 ng/L (plasma) 5.9–6.5% at 22 ng/L (whole blood)	6.5% (plasma) 6.1% at 22.9 ng/L (whole blood)
Concentration at 20% CV	4 ng/L	2.1–3.6 ng/L (plasma) 2.8 ng/L (whole blood)	2.1 ng/L (plasma) 3.7 ng/L (whole blood)
Concentration at 10% CV	15 ng/L	4.4–8.4 ng/L (plasma) 5.8–6.2 ng/L (whole blood)	6.7 ng/L (plasma) 8.9 ng/L (whole blood)
99th Percentile	Overall: 27.9 ng/L F: 20.3 ng/L M: 29.7 ng/L	Overall: 20.5 ng/L F: 14.4 ng/L M: 25.7 ng/L	Overall: 22.9 ng/L F: 18.5 ng/L M: 27.1 ng/L
Reference population	Overall n = 734 F: 352 M: 382	Overall n = 789 F: 391 M: 398	Overall n = 694 F: 331 M: 363
Specimen type	Heparin-Na, heparin-Li or EDTA whole blood or plasma	EDTA whole blood or plasma	Li Hep whole blood and plasma, capillary blood
POCT definition (according to Collinson 2023)	Desktop	Portable	Portable
Instrument size	475 × 343 × 569 mm	190 × 70 × 225 mm	250 × 52 × 85 mm (analyzer) 290 × 60 × 100 mm (docking station)
Instrument weight	28 kg	0.7 kg	780 g (analyzer) 460 g (docking station)
Single-test technology	No	Yes	Yes
Other markers available on the same analyzer	NT-proBNP, myoglobin, CK-MB, D-dimer, presepsin, CRP, PCT	BNP, NT-proBNP, myoglobin, CK-MB, D-dimer, drugs of abuse, PLGF	No
Included in ESC 2023 guidelines for ACS [74]	Yes	Yes	No

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
