# Peer review of "Circulating Biomarkers for Monitoring Chemotherapy-Induced Cardiotoxicity in Children"

_pharmaceutics, 2023, doi:10.3390/pharmaceutics15122712_

Round 1
Reviewer 1 Report
Comments and Suggestions for Authors
The manuscript entitled: “Chemotherapy-induced cardiotoxicity and the emerging use of high-sensitivity troponin as a biomarker in pediatric patients” fits well into the aims and scope of Pharmaceutics.
Comments:
The manuscript handles an important point and is well-written however, some corrections are required:
- Some abbreviations are missed so please add a list of abbreviations.
-Page 3 lines 50-63, this paragraph without reference. Please add a reference.
-Page 5 Line 135 do not start with “are….” So please add Anthracyclines first
- Page 5, section 2.1. Anthracyclines, it is too long paragraph from lines 134-222.
- Page 6, line 196 “Tarek Magdy and colleagues” please correct the name to be Magdy ….. and add the reference number.
-In section 2.3. Tyrosine kinase inhibitors, please don’t start with the abbreviation “(TKIs) block the enzymes tyrosine kinases.”
- In section 2.4. Alkylating agents, please do not start with “such as…”
-Page 8 lines 268-271, please add a reference “
Comments on the Quality of English LanguageThe English language is fine.
Author Response
The manuscript entitled: “Chemotherapy-induced cardiotoxicity and the emerging use of high-sensitivity troponin as a biomarker in pediatric patients” fits well into the aims and scope of Pharmaceutics.
Comments:
The manuscript handles an important point and is well-written however, some corrections are required:
- Some abbreviations are missed so please add a list of abbreviations.
We add a list of abbreviations after Conclusion section.
-Page 3 lines 50-63, this paragraph without reference. Please add a reference.
We add references in lines 49-51 and at the end of paragraph.
-Page 5 Line 135 do not start with “are….” So please add Anthracyclines first
We fixed the typo.
- Page 5, section 2.1. Anthracyclines, it is too long paragraph from lines 134-222.
According to reviewer suggestion, we have divided the paragraph introducing a sub-paragraph entitled “Genetic variants associated with Anthracycline sensitivity”
- Page 6, line 196 “Tarek Magdy and colleagues” please correct the name to be Magdy ….. and add the reference number.
We modified the reference according to the reviewer suggestion.
-In section 2.3. Tyrosine kinase inhibitors, please don’t start with the abbreviation “(TKIs) block the enzymes tyrosine kinases.”
We fixed the typo.
- In section 2.4. Alkylating agents, please do not start with “such as…”
We fixed the typo.
-Page 8 lines 268-271, please add a reference “
We add a reference.
Reviewer 2 Report
Comments and Suggestions for Authors
The title of this review is dedicated to the use of high sensitivity troponin for the detection of cardiotoxicity.
However, there are no data supporting this assessment, in pediatric cancer.
This review is mainly devoted to adult cardiotoxicity: for example, taxane are not used in pediatric cancer.
It is stated that treatments of pediatric cancer are derivated from adult ones; that is not true and the toxic limited doses of agents of chemotherapies are well known in children. There is few cardiac long term toxicities except for children who experiences relapses and second line treatments.
The genetic aspects of predisposition are interesting.
This review is interesting but the title should be changed. If High sensibility troponin might be an interesting tool for early detection, that remains to be prooved by prospective studies, particularly in childhood, but also in adulthood.
Author Response
The title of this review is dedicated to the use of high sensitivity troponin for the detection of cardiotoxicity.
However, there are no data supporting this assessment, in pediatric cancer.
- We agree that the title “Chemotherapy-induced cardiotoxicity and the emerging use of high-sensitivity troponin as a biomarker in pediatric patients” may be inappropriate since there are still few preliminary studies in children describing the use of hsTnI (see references113-117). So, according to the reviewer suggestion, we propose a new title: “Circulating biomarkers for monitoring chemotherapy-induced cardiotoxicity in children”.
This review is mainly devoted to adult cardiotoxicity: for example, taxanes are not used in pediatric cancer.
It is stated that treatments of pediatric cancer are derivated from adult ones; that is not true and the toxic limited doses of agents of chemotherapies are well known in children. There is few cardiac long term toxicities except for children who experiences relapses and second line treatments.
- According to reviewer comment we rephrased the sentence (lines 97-104) better describing how chemotherapy regimen are established in children.
- We add a new sentence in paragraph 2 (lines 129-130) to specify that long term cardiotoxicity regard children who experiences relapses and second line treatments as rightly suggested by reviewer.
The genetic aspects of predisposition are interesting.
- We sincerely thank the reviewer for this appreciation.
This review is interesting but the title should be changed. If High sensibility troponin might be an interesting tool for early detection, that remains to be proved by prospective studies, particularly in childhood, but also in adulthood.
- We have proposed a new title as shown above.
- According to reviewer consideration, we updated the table 2 with POC systems that are suitable for pediatric patients as they require small volume of whole blood samples (see lines 368-373). In addition, we add in Conclusion section (lines 518-520) a sentence to further underline, as rightly suggested by the reviewer, that high sensitivity troponin is a very interesting tool for early detection of cardiotoxicity, but future prospective studies, in childhood and in adulthood are required
Reviewer 3 Report
Comments and Suggestions for Authors
This narrative review provides an exhaustive analysis of the impact of various chemotherapeutic agents on cardiac function, with a particular focus on the primary biomarkers utilized to detect and monitor cardiotoxicity. It emphasizes the pivotal role that high-sensitivity troponin plays as a biomarker in tracking chemotherapy-induced cardiotoxicity in pediatric patients.
The review stands as a robust contribution to the current body of literature and is augmented by well-designed figures and tables for enhanced comprehension and presentation.
Regarding the visual abstract, it is imperative that it stands on its own as a succinct and informative summary. To this end, I recommend it be revised to more effectively illustrate the "Hallmarks of using high-sensitivity troponin as a biomarker for chemotherapy-induced cardiotoxicity in pediatric patients".
A visual abstract should serve as a quick, visual synopsis of the paper's core message, designed to facilitate immediate understanding, foster cross-disciplinary engagement, and help researchers swiftly pinpoint the papers most pertinent to their areas of interest. Consequently, a revision aimed at clarifying its presentation would be highly beneficial.
Line 196: please correct "Tarek Magdy and colleagues" and remove the underline
Line 449: please delete the extra space
Line 453: please replace "Sars-Cov-2" with SARS-CoV-2
Author Response
This narrative review provides an exhaustive analysis of the impact of various chemotherapeutic agents on cardiac function, with a particular focus on the primary biomarkers utilized to detect and monitor cardiotoxicity. It emphasizes the pivotal role that high-sensitivity troponin plays as a biomarker in tracking chemotherapy-induced cardiotoxicity in pediatric patients.
The review stands as a robust contribution to the current body of literature and is augmented by well-designed figures and tables for enhanced comprehension and presentation.
Regarding the visual abstract, it is imperative that it stands on its own as a succinct and informative summary. To this end, I recommend it be revised to more effectively illustrate the "Hallmarks of using high-sensitivity troponin as a biomarker for chemotherapy-induced cardiotoxicity in pediatric patients".
A visual abstract should serve as a quick, visual synopsis of the paper's core message, designed to facilitate immediate understanding, foster cross-disciplinary engagement, and help researchers swiftly pinpoint the papers most pertinent to their areas of interest. Consequently, a revision aimed at clarifying its presentation would be highly beneficial.
We thank the reviewer to give us the opportunity to improve the graphical abstract. In order to effectively illustrate the “Hallmarks of using high-sensitivity troponin as a biomarker for chemotherapy-induced cardiotoxicity in pediatric patients", as he suggested, we introduced into the scheme a hint to the high sensitivity Troponin method underlying its advantages. We hope this new version of the figure satisfied the reviewer’s request.
Line 196: please correct "Tarek Magdy and colleagues" and remove the underline
We corrected the reference ad removed the underline.
Line 449: please delete the extra space
We deleted the extra-space.
Line 453: please replace "Sars-Cov-2" with SARS-CoV-2
We fixed the typo.
Round 2
Reviewer 2 Report
Comments and Suggestions for Authors
OK